# Crosstalk between guanosine nucleotides regulates cellular heterogeneity in protein synthesis during nutrient limitation

**Simon Diez, Molly Hydorn**[ID]**, Abigail Whalen**[ID]**, Jonathan Dworkin**[ID]*

Department of Microbiology and Immunology, College of Physicians and Surgeons, Columbia University, New York, New York, United States of America

* jonathan.dworkin@columbia.edu

**Data Availability Statement:** All relevant data are within the manuscript and its Supporting Information files.

## Abstract

Phenotypic heterogeneity of microbial populations can facilitate survival in dynamic environments by generating sub-populations of cells that may have differential fitness in a future environment. *Bacillus subtilis* cultures experiencing nutrient limitation contain distinct sub-populations of cells exhibiting either comparatively high or low protein synthesis activity. This heterogeneity requires the production of phosphorylated guanosine nucleotides (pp)pGpp by three synthases: SasA, SasB, and RelA. Here we show that these enzymes differentially affect this bimodality: RelA and SasB are necessary to generate the sub-population of cells exhibiting low protein synthesis whereas SasA is necessary to generate cells exhibiting comparatively higher protein synthesis. Previously, it was reported that a RelA product allosterically activates SasB and we find that a SasA product competitively inhibits this activation. Finally, we provide *in vivo* evidence that this antagonistic interaction mediates the observed heterogeneity in protein synthesis. This work therefore identifies the mechanism underlying phenotypic heterogeneity in protein synthesis.

## Author summary

Upon encountering conditions unfavorable to growth such as nutrient limitation, bacteria enter a quiescent phenotype that is mediated by group of guanosine nucleotides collectively known as (pp)pGpp. These nucleotides direct the down-regulation of energy intensive processes and are essential for a striking heterogeneity in protein synthesis observed during exit from rapid growth. Here, we show that a network of (pp)pGpp synthases is responsible for this heterogeneity and describe a mechanism that allows for the integration of multiple signals into the decision to down regulate the most energy intensive process in a cell.

**Funding:** SD was supported in part by the Columbia University Graduate Training Program in Microbiology, Immunology and Infection (NIH, R01 AI106711). JD was supported by NIH R01GM141953, R35GM141953, R21AI156397, and is a Burroughs-Welcome Investigator in the Pathogenesis of Infectious Disease (#1010084). The funders had no role in study design, data collection and analysis, decision to publish, or preparation of the manuscript.

**Competing interests:** The authors have declared that no competing interests exist.

## Introduction

Nutrient availability is a major environmental cue for bacteria. For example, amino acid starvation results in induction of the stringent response, a conserved mechanism dependent on the synthesis of the nucleotides guanosine penta- and tetra-phosphate ((p)ppGpp). These nucleotides mediate a broad shut down of energy intensive reactions which are required during rapid growth [1,2]. (p)ppGpp directly binds and inhibits key proteins that catalyze processes including transcription (RNA polymerase [3,4]), translation (GTPase IF2 [5]), GTP biosynthesis (HprT and GmK [6]), DNA replication (DNA primase [7]), and ribosome assembly (ObgE [8]).

Gram-positive bacteria typically encode a single, bi-functional RSH (RelA-SpoT Homolog) enzyme capable of both (p)ppGpp synthesis and hydrolysis as well as two additional small alarmone synthases (SAS) that lack hydrolytic activity. Unlike RSH proteins, which are activated by the binding of a deacylated tRNA in the A-site of the ribosome, SAS enzymes are often transcriptionally regulated [9] and some are also under allosteric control [10]. RelA/SpoT and the SAS synthases preferentially produce different molecules in different species. For example, in response to amino acid starvation, *E. coli* RelA produces approximately equal amounts of the tetra-phosphorylated (ppGpp) and the penta-phosphorylated (pppGpp) guanosines, whereas *B. subtilis* RelA primarily generates pppGpp using GTP and ATP as substrates [11]. *B. subtilis* SasB preferentially utilizes GDP and ATP to generate the tetra-phosphorylated guanosine (ppGpp) [12]. SasA, the other SAS enzyme in *B. subtilis*, generates either a 5' monophosphate 3' di-phosphate guanosine or a 5' di-phosphate 3' monophosphate (pGpp) [12], at least in part due to the action of the NahA hydrolase [13]. Together, these three closely related nucleotides are referred to as (pp)pGpp.

Recently, our laboratory demonstrated that accumulation of (pp)pGpp attenuates protein synthesis when populations of *B. subtilis* cease growing exponentially [5]. This attenuation is bimodal and results in a heterogeneity in the protein synthesis activity of individual cells that exhibit either comparatively high or low protein synthesis activity [5]. Here we find that all three *B. subtilis* (pp)pGpp synthases including the RSH protein RelA and SAS proteins SasA and SasB are required for this heterogeneity since the absence of any of these synthases results in the loss of bimodality. The SasA product pGpp and the RelA product pppGpp together antagonistically regulate activation of the third synthase (SasB), that is itself responsible for the synthesis of ppGpp, a molecule that directly inhibits the initiation of translation during nutrient limitation [5].

## Results

### The SasA and SasB (pp)pGpp synthases contribute to heterogeneity

Cellular heterogeneity in protein synthesis as *B. subtilis* cultures exit rapid growth is dependent on the presence of the phosphorylated guanosine nucleotides (pp)pGpp [5]. We investigated the origins of this heterogeneity by assessing single cell protein synthesis using *O*-propargyl-puromycin (OPP) incorporation in strains carrying deletion mutations in either of the two *B. subtilis* (pp)pGpp synthases (SasA and SasB) whose expression increases during exit from rapid growth [12]. To quantify these effects we applied a cutoff that specifies the population of cells with low rates of protein synthesis. Nearly all cells of a *B. subtilis* stationary phase culture exhibit very low protein synthesis [5] so we defined this cutoff (850 arbitrary fluorescence units (au)) as the magnitude of OPP labeling of a *B. subtilis* culture in stationary phase that captures >95% of the entire population (S1 Fig). We used this threshold to define the fraction

of the population with low rates of protein synthesis during late transition phase ($OD_{600}$ ~0.685) as "OFF" (S2 Fig). By convention, we define the remainder of the population as "ON."

A strain lacking SasB (*ΔsasB*) contained fewer "OFF" cells as compared to the wildtype strain (Fig 1A and 1B). This result is consistent with our previous observation that the SasB product ppGpp inhibits the function of IF2 and thereby downregulates protein synthesis [5]. In striking contrast, a strain lacking SasA (*ΔsasA*) did not contain the substantial fraction of "ON" cells seen in the wildtype parent strain (Fig 1A and 1C) and most cells in the population were "OFF". This observation suggests that the SasA product pGpp does not inhibit translation, as does the SasB product ppGpp. Consistently, unlike ppGpp, pGpp does not directly bind known translational GTPases (e.g., EF-G [14])

## *sasA* but not *sasB* expression is correlated with levels of protein synthesis

*sasA* and *sasB* are regulated transcriptionally and expressed post-exponentially [12,15] when the heterogeneity is observed (Fig 1A). We therefore asked if expression of either *sasA* or *sasB* is correlated with protein synthesis using transcriptional fusions of the *sasA* or the *sasB* promoters to the gene encoding YFP ($P_{sasA}$-*yfp* or $P_{sasB}$-*yfp)*. Consistent with prior observations [12], expression of both *sasA* and *sasB* reporters increased during the exit from exponential growth (Fig 2A and 2B). We examined the relationship between promoter activity and protein synthesis by measuring both YFP expression and OPP incorporation in single cells. Cells with higher *sasA* expression ($P_{sasA}$-*yfp)* were more likely to have higher levels of protein synthesis than cells with lower *sasA* expression (Fig 2D). If the population is divided into quartiles of *sasA* expression, average OPP incorporation in the top two quartiles as compared to the bottom quartile was significantly higher (Fig 2D). In comparison, the difference in OPP incorporation between any of the quartiles of *sasB* expression (Fig 2C) was not significant. Thus, differences in *sasA*, but not *sasB*, expression are associated with the observed heterogeneity in protein synthesis.

## SasB allosteric activation is necessary for heterogeneity

If changes in *sasB* transcription are not associated with differences in protein synthesis (Fig 2C), but SasB is necessary for the heterogeneity of protein synthesis (Fig 1B), what mechanism underlies differential SasB activity in single cells? *B. subtilis* SasB is subject to allosteric activation by pppGpp, the main product of *B. subtilis* RelA [16]. Phe-42 is a key residue in this activation and a SasB mutant protein carrying an F42A substitution (SasB[F42]) is not allosterically activated by pppGpp *in vitro* [16]. We investigated the importance of this allosteric activation for protein synthesis heterogeneity using a strain expressing SasB[F42] instead of SasB. Since heterogeneity of this strain was significantly attenuated compared to the WT strain (Fig 3A and 3B), allosteric activation of SasB by pppGpp is key for the bimodality of protein synthesis activity.

This result suggests that the enzyme responsible for pppGpp synthesis could also affect the heterogeneity. RelA is the primary source of pppGpp in *B. subtilis* [11], so loss of *relA* would be predicted to affect SasB activity. We therefore generated a strain expressing a RelA mutant protein (RelA[Y308A]) carrying a single amino acid change at a conserved residue essential for synthase but not hydrolysis activity [17,18] since RelA hydrolytic activity is essential in a strain that retains functional *sasA* and *sasB* genes [19]. Labeling of this strain with OPP in late transition phase revealed that the "OFF" population was largely absent (Fig 3C), demonstrating that RelA-mediated pppGpp synthesis is important for the bimodality.

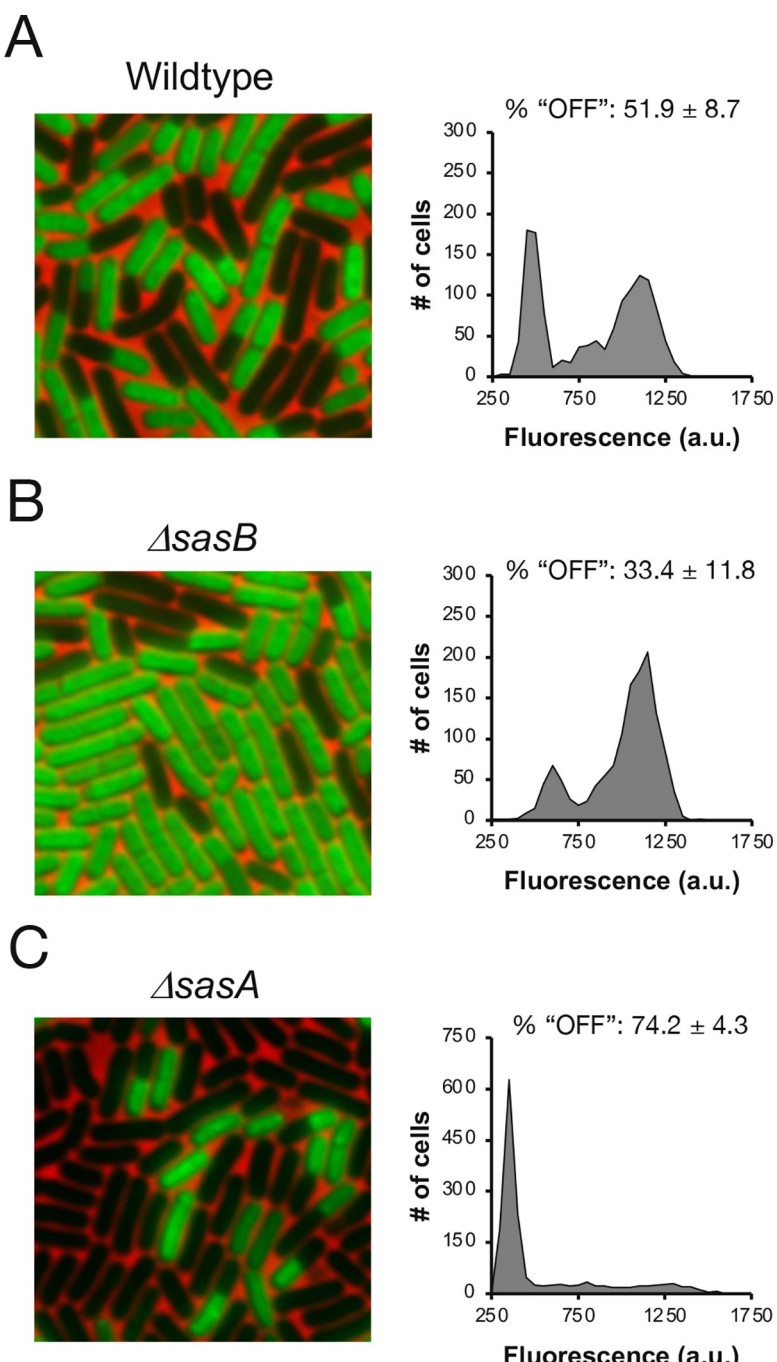

**Fig 1. *sasB* and *sasA* have opposite effects on bimodality. (A, B, C)** Representative pictures and population distributions of OPP labeled **(A)** wildtype (JDB1772), **(B)** Δ*sasB* (JDB4310) and **(C)** Δ*sasA* (JDB4311) during late transition phase. % of population "OFF" as determined in S1 Fig presented above each representative distribution. Statistical significance was determined by comparing three independent populations of WT to either mutant. P-values are 0.046 and 0.011 for Δ*sasB* and Δ*sasA* respectively.

## SasB allosteric activation is inhibited by pGpp

A strain lacking SasA (Δ*sasA*) contains more "OFF" cells as compared to the wildtype parent (Fig 1C). The presence of this sub-population of cells depends on a SasB protein that can be

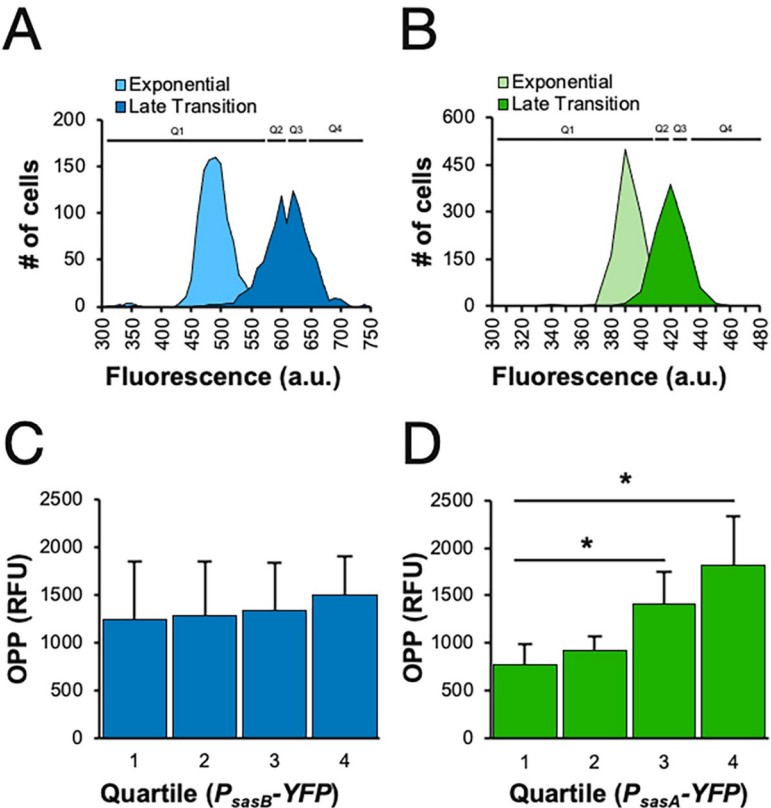

**Fig 2. Relationship between *sasA* or *sasB* expression and OPP incorporation. (A, B)** Representative population distribution of *B. subtilis* carrying a transcriptional reporter of **(A)** $P_{sasB}$-*yfp* (JDB4341) or **(B)** $P_{sasA}$-*yfp* (JDB4030) during exponential (light blue/green) and late transition phase (dark blue/green). Black lines represent quartiles used in C and D. **(C, D)** Average OPP incorporation in late transition phase of each quartile of **(C)** $P_{sasB}$-*yfp* expression or **(D)** $P_{sasA}$-*yfp* expression from lowest to highest. Statistical analysis (one tailed t-test) showed no significant difference in OPP incorporation between any $P_{sasB}$-*yfp* quartiles (p>0.05) and significantly higher OPP incorporation between quartiles 1 and 3 and quartiles 1 and 4 of $P_{sasA}$-*yfp* expression (p-values 0.027 and 0.016, respectively).

allosterically activated (Fig 3B). Integrating these two observations, we hypothesized that a product of SasA (pGpp) inhibits the allosteric activation of SasB by pppGpp. pGpp and pppGpp could have an antagonistic interaction since they are likely capable of binding to the same site on SasB, but their differing phosphorylation states could affect their ability to allosterically activate SasB.

We tested this possibility by assaying *in vitro* whether pGpp inhibits the allosteric activation of SasB. First, we confirmed that SasB generates more ppGpp when reactions are supplemented with pppGpp and, as reported [16], we observed a ~2 fold increase in ppGpp production when SasB was incubated with pppGpp (Fig 4A). Using pGpp synthesized *in vitro* by the (pp)pGpp hydrolase NahA [14], we observed that pGpp attenuates the allosteric activation of SasB in a dose dependent manner (Fig 4A). Since even the highest concentration of pGpp did not decrease production of ppGpp relative to that generated by SasB without the addition of pppGpp (Fig 4A), the inhibition is likely specific to the allosteric activation. We tested this directly by assaying the effect of pGpp on SasB activity in the absence of its allosteric activator (pppGpp). Addition of pGpp did not significantly affect SasB activity within the range of pGpp concentrations we used previously (S3 Fig). We further confirmed the specificity by assaying a SasB[F42] mutant protein that is insensitive to allosteric activation by pppGpp [16]. As reported SasB[F42A] has similar activity to a non-allosterically activated WT SasB in the presence

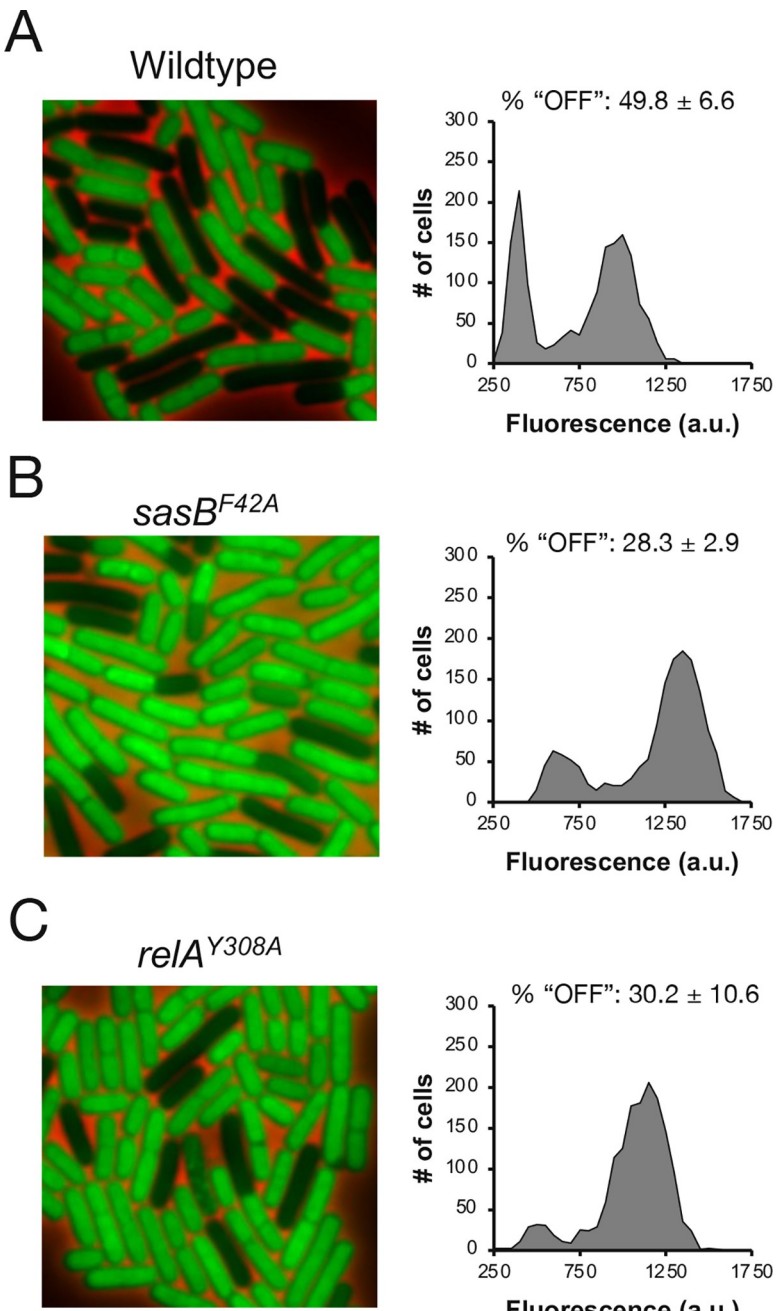

**Fig 3. Allosteric activation of SasB is required for bimodality during exit from rapid growth. (A, B,C)**
Representative pictures and population distributions of OPP labeled (**A**) wildtype (JDB1772), (**B**) *sasB*<sup>F42A</sup> (JDB4340), and (**C**) *relA*<sup>Y308A</sup> (JDB4300) strains during late transition phase. % of population "OFF" as determined in S1 Fig is presented above each representative distribution. Statistical significance was determined by comparing 3 independent populations of WT to either mutant. P-values are 0.002 and 0.035 for *sasB*<sup>F42A</sup> and *relA*<sup>Y308A</sup> respectively.

of pppGpp (Fig 4B). However, in contrast with wildtype SasB, pGpp does not affect the activity of SasB<sup>F42A</sup> even when pppGpp is included (Fig 4B).

These *in vitro* biochemical experiments suggest that the effect of SasA on protein synthesis heterogeneity is dependent on the activity of SasB. If this is true *in vivo*, then a Δ*sasB* mutation

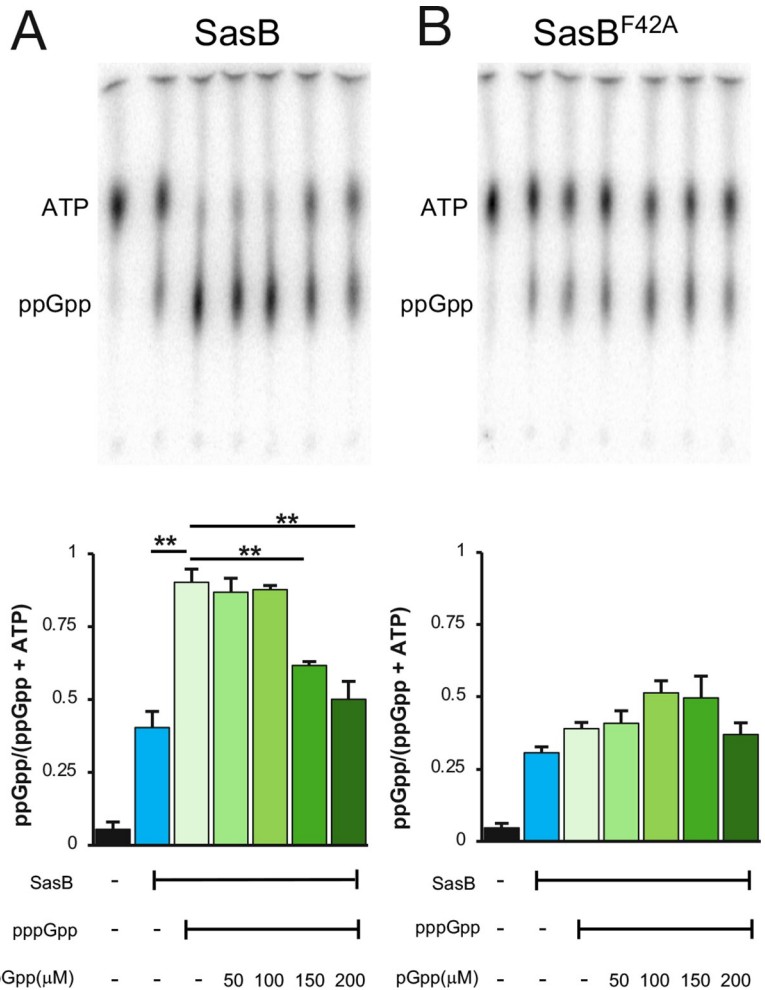

**Fig 4. pGpp inhibits the allosteric activation of SasB by pppGpp. (A)** Representative TLC of nucleotides present following incubation of wildtype SasB with $[\alpha\text{-}^{32}\text{P}]$-ATP and GDP in the presence or absence of pppGpp and increasing concentrations of pGpp ($\mu$M) (top). Quantitation of the ratio of ppGpp to total nucleotides present in each lane in TLC. This ratio was calculated using the formula: ppGpp/ATP + ppGpp (bottom). **(B)** Representative TLC of nucleotides present following incubation of SasB$^{F42A}$ with $[\alpha\text{-}^{32}\text{P}]$-ATP and GDP in the presence or absence of pppGpp and increasing concentrations of pGpp (top). Ratio of ppGpp present in each lane in TLC as determined the formula, ppGpp/ATP + ppGpp (bottom). Statistical analysis (two tailed t-test) showed no significance (p>0.05) between reactions containing SasB in the presence or absence of pppGpp and/or pGpp.

should be dominant to a $\Delta sasA$ mutation. Consistently, the population of "OFF" cells in a $\Delta sasA$ strain was absent in a strain lacking both SasA and SasB ($\Delta sasA\ \Delta sasB$) (Fig 5A and 5B). Thus, the effect of SasA is dependent *in vivo* on SasB. Finally, since RelA activates SasB, a *relA* mutation should be dominant to a $\Delta sasA$ mutation. Consistently, a strain expressing RelA$^{Y308A}$ and carrying a $\Delta sasA$ mutation exhibited a loss of heterogeneity in protein synthesis similar to the *relA*$^{Y308A}$ strain, demonstrating that the effect of the $\Delta sasA$ mutation depends on a functional RelA synthase (Fig 5A and 5C). Since pGpp also accumulates in stationary phase cells as a result of degradation of both ppGpp and pppGpp by the hydrolase NahA [14,20], we asked if NahA contributes to the heterogeneity in protein synthesis by comparing OPP incorporation in wildtype and $\Delta nahA$ cells during late transition phase. Since we observed no difference in heterogeneity (S4 Fig), SasA is the primary regulator of heterogeneity under our experimental conditions.

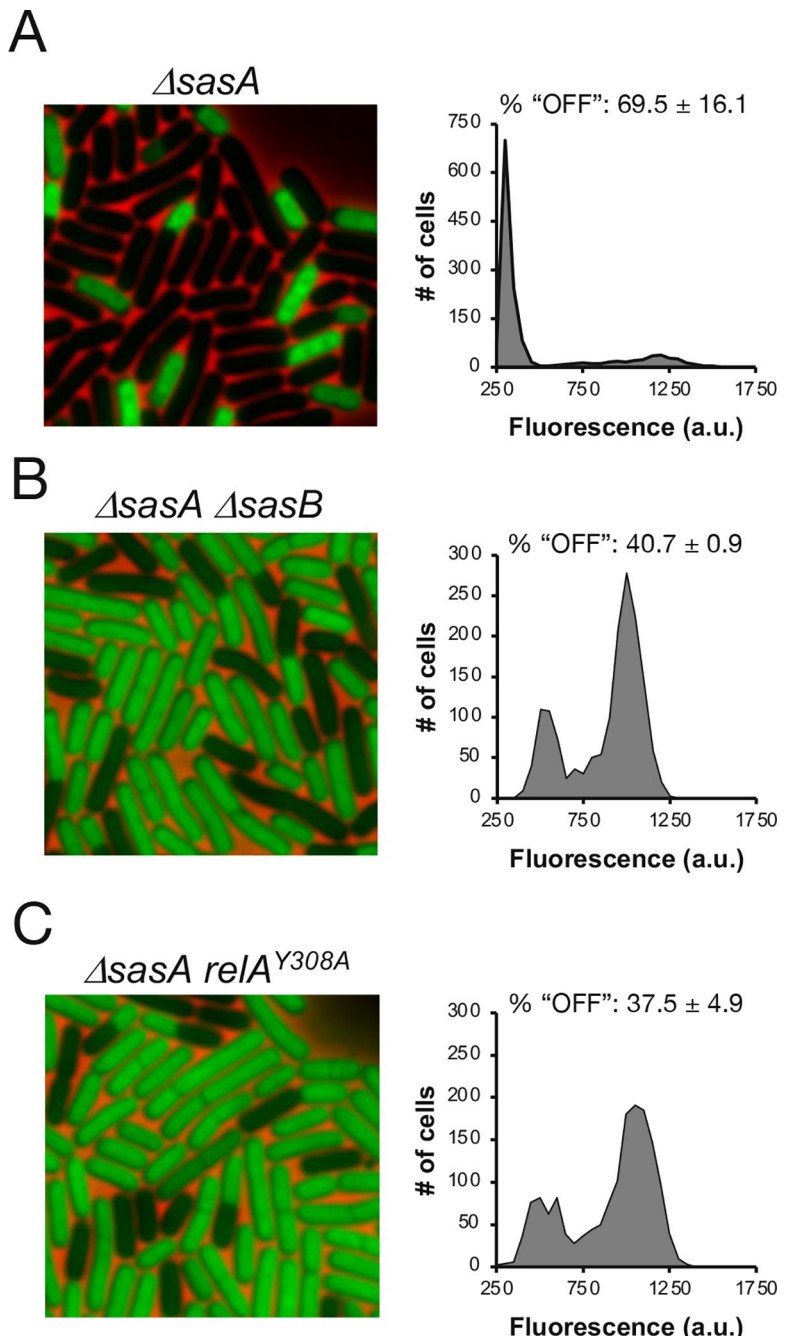

**Fig 5. *sasA* effect is dependent on *sasB* and *relA*. (A, B, C)** Representative pictures and population distributions of OPP labeled **(A)** *ΔsasA* (JDB4310), **(B)** *ΔsasA ΔsasB* (JDB4312) **(C)** *ΔsasA relA*^Y308A^ (JDB 4301) strains during late transition phase. % of population "OFF" as defined in S1 Fig is presented above each representative distribution. Statistical significance was determined by comparing 3 independent populations of WT to either mutant. P-values are 0.028 and 0.030 for *ΔsasAΔsasB* and *ΔsasA relA*^Y308A^, respectively.

## Discussion

*B. subtilis* populations experiencing nutrient limitation and entering into quiescence respond bimodally with respect to global protein synthesis activity [5]. Here, we find that this

bimodality depends on all three (pp)pGpp synthases. We demonstrate that it is dependent on the allosteric activation of SasB by the RelA product pppGpp and that this activation is antagonized by a SasA product pGpp. Our work therefore provides a mechanism for the phenotypic heterogeneity observed and identifies a novel regulatory interaction between (pp)pGpp synthases.

## Regulation of protein synthesis during nutrient limitation

The down-regulation of protein synthesis in *B. subtilis* cells experiencing nutrient limitation occurs in part as a result of ppGpp directly inhibiting IF2 [5]. SasB is the main source of ppGpp and this work identifies how ppGpp synthesis by SasB and the subsequent downregulation of protein synthesis is coupled to changes in environmental conditions. First, SasB allosteric activation by the RelA product pppGpp is required for the downregulation of protein synthesis in a subpopulation of cells (Figs 3 and 5). RelA activity reflects tRNA charging levels [11], thereby coupling SasB-dependent regulation of protein synthesis to amino acid availability. Second, a SasA product (pGpp) inhibits the allosteric activation of SasB (Fig 4). Although SasA is constitutively active, *sasA* expression, at least in part, reflects availability of the Lipid II peptidoglycan precursor [21–23], thereby coupling SasB-dependent regulation of protein synthesis to cell wall metabolism. Thus, RelA and SasA dependent regulation of SasB may integrate multiple environmental signals in the decision to attenuate protein synthesis.

## Physiological sources of variability in SasB activity

Phenotypic heterogeneity such as that observed here in the context of protein synthesis can arise from stochastic differences in gene expression [24]. Although *sasB* expression exhibits substantial variability in expression cell to cell (Fig 2A), it does not correspond with the level of protein synthesis in individual cells (Fig 2C). Thus, variability of SasB activity in single cells is likely relevant. What could be responsible? Our observations link heterogeneity to the convergent regulation of SasB allosteric activation by the products of the RelA and SasA synthases (Fig 4A). Thus, both enzymes are potential sources of variability and, consistently, strains carrying either *relA*^*Y308A* or Δ*sasA* mutations exhibit a loss in heterogeneity as compared to the wildtype (Figs 1C and 3C). Since RelA is a cellular sensor of tRNA charging, levels of which are highly sensitive to growth conditions [25], variations in this parameter could contribute to variability in protein synthesis via modulation of RelA activity. Noise in *sasA* transcription is dependent on the activity of PrkC, a membrane Ser/Thr kinase that regulates *sasA* via the essential WalRK two component system [21]. Since both WalRK [26] and PrkC [27] activities reflect cell wall metabolism, variation in this process could also impact *sasA* variability. Thus, differences in the protein synthesis activity of individual cells may reflect cellular variations in amino acid and/or cell wall metabolism.

## Allosteric activation of (pp)pGpp synthases

Many genes encoding SAS proteins such as *sasB* are transcriptionally regulated [9]. In addition, we observe here that allosteric activation of SasB by pppGpp [16] is required for the attenuation of protein synthesis (Fig 3) demonstrating that *sasB* transcription is necessary but not sufficient, at least in the physiological context of nutrient limitation. We also find that this allosteric activation is antagonized by a SasA product pGpp, consistent with the epistatic relationship between *sasB* and *sasA* (Fig 5A). Antagonistic regulatory mechanisms are likely widespread in this family of synthases. For example, the SasB homolog *Enterococcus faecalis* RelQ is attenuated by RNA that competes with pppGpp for binding to the allosteric site [28]. Given the very recently observed allosteric activation of *B. subtilis* RelA by (p)ppGpp [29], an

important question for future study is to determine whether this activation is also subject to antagonism by pGpp and, if so, to characterize the physiological consequences of this regulation.

## (pp)pGpp synthases

Strains carrying single mutations in one of the three genes encoding a (pp)pGpp synthase (ΔsasA, ΔsasB, relA; Figs 1 and 3C) exhibit different protein synthesis activity, consistent with previous reports that SAS enzymes differ between themselves and also with RelA in the guanosine nucleotide that they preferentially produce [14,30–32]. Our experiments thereby extend previous observations that ppGpp and pppGpp can differ in their effect on gene transcription in *E. coli* [33]. The biochemical experiments demonstrating that pGpp antagonizes pppGpp allosteric activation of SasB, but itself is not capable of activation (Figs 4A and S3) are consistent with our physiological experiments. The biochemical activity of these nucleotides have been reported to differ, including observations that pppGpp is much more potent than ppGpp in stimulating SasB [16], that pGpp is a significantly more potent inhibitor of purine salvage enzyme XPRT than ppGpp [34], and that ppGpp, but not pppGpp, inhibits the function of IF2 in stimulating subunit joining [35].

## Physiological implications of heterogeneity in protein synthesis

(pp)pGpp has long been thought to mediate entry into bacterial quiescence [36,37]. This transition facilitates survival in nutrient limited environments and its regulation depends upon the integration of a multitude of rapidly changing environmental signals that themselves may impair decision-making. One way bacteria deal with such uncertainty is to generate subpopulations, with distinct, often bimodal phenotypes from a population of genetically identical cells [24]. Examples of phenotypic variation in *B. subtilis* include heterogeneity in specific metabolic activities such as acetate production [38] or in developmental transitions such as sporulation [39] and competence [40]. The phenotypic variation in protein synthesis activity we observe here has potentially important functional implications. A global reduction in protein synthesis activity, if accompanied by a constant rate of protein degradation, would have the effect of reducing overall metabolic capacity, especially by affecting processes like ribosome assembly. Global effects also could have specific regulatory consequences. For example, the alternative sigma factor *B. subtilis* SigD drives expression of genes controlling daughter cell separation and motility that exhibit well characterized phenotypic variation. RelA affects both this variability as well as absolute levels of SigD [41], suggesting that differences in protein synthesis between cells may contribute to SigD variability.

In summary, this work demonstrates that the three (pp)pGpp synthases comprise a signaling network responsible for the heterogenous regulation of protein synthesis as *B. subtilis* cultures enter quiescence. We find that this heterogeneity is dependent on the RelA product pppGpp, which allosterically activates SasB, and a SasA product, pGpp, which antagonizes this activation. Since the activities of RelA and SasA reflect amino acid and peptidoglycan precursor availability, respectively, these parameters are thereby coupled to protein synthesis activity and facilitate cell decision making during the entry into quiescence.

## Materials and methods

### Strains and media

Strains were derived from *B. subtilis* 168 *trpC2*. *sasA* (*ywaC*) and *sasB* (*yjbM*) gene knockouts were from transformed into *B. subtilis* 168 *trpC2* using genomic DNA from BD5467 [42]. The

*sasB* transcriptional reporter strain was constructed similarly as described [21]. Briefly, a 107 bp region encompassing the *sasB* operon promoter ($P_{sasB}$) was amplified and inserted into AEC 127 using *EcoR*I and *BamH*I sites. The resulting AEC 127 $P_{sasB}$ was integrated into *B. subtilis* 168 *trpC2* at *sacA*. *sasB*$^{F42A}$ and *relA*$^{Y308A}$ strains were generated using integration of pMINIMAD2 derivatives (pMINIMAD2 *sasB*$^{F42A}$ and pMINIMAD2 *relA*$^{Y308A}$, respectively). Briefly, *sasB* was amplified excluding start and stop codons and F42A mutation was introduced using overlap extension PCR. *sasB*$^{F42A}$ was inserted into pMINIMAD2 vector using *EcoR*I and *Sal*I sites. pMI-NIMAD2 *sasB*$^{F42A}$ vector was transformed into *B. subtilis* 168 *trpC2* using a standard transformation protocol. Transformants were selected for erythromycin resistance at 45˚C overnight and grown for 8 hours at RT in LB. Cultures were diluted 1:10 in LB and grown overnight. Cultures were plated for single colonies and grown overnight at 45˚C. Single colonies were checked for erythromycin sensitivity and sensitive clones were checked for *sasB*$^{F42A}$ allele by Sanger sequencing of *sasB* amplified genomic region. The *relA*$^{Y308A}$ strain was generated in a similar way but *EcoR*I and *BamH*I sites were used to insert the *relA*$^{Y308A}$ gene into pMINIMAD2.

## Growth curves

Growth curves were performed in a Tecan Infinite m200 plate reader at 37˚C with continuous shaking and $OD_{600}$ measurements were made every five min. Cultures were grown from single colonies from fresh LB plates grown overnight at 37˚C. Exponential phase starter cultures ($OD_{600} \sim 0.5–1.5$) were diluted to $OD_{600} = 0.01$ and grown in 96-well Nunclon Delta surface clear plates (Thermo Scientific) with 150 μL per well. All growth curves were done in triplicate and media-only wells were used to subtract background absorbance.

## OPP labeling

OPP labeling of cells was as described [5]. Exposure times were 30 msec for phase contrast, and 20 msec for mCherry. Fluorescence intensity of ~1570 single cells per experiment was determined using ImageJ. Cells were binned based on fluorescence intensity using 50 a.u. wide bins in all experiments and number of cells in each bin presented as a histogram.

## Protein expression and purification

Wildtype and F42A SasB proteins were expressed and purified essentially as described [16]. Wildtype *sasB* was amplified from *B. subtilis* 168 *trpC2*. The F42A mutation was introduced using overlap extension PCR. WT and *sasB*$^{F42A}$ PCR products were inserted into pETPHOS expression vector using *EcoR*I and *BamH*I sites. pETPHOS WT *sasB* and pETPHOS *sasB*$^{F42A}$ were transformed into *E.coli* BL21 and proteins were induced with 1 mM IPTG for 2h at $OD_{600} \sim 0.5$. Cells were harvested at 4˚C and lysed using a Fastprep (MP biomedicals) in 50 mM Tris (pH 8.0), 250 mM NaCl, 5 mM $MgCl_2$, 2 mM BME, 0.2 mM PMSF, and 10mM imidazole. Lysates were clarified and bound to a Ni-NTA column (Qiagen) for 1h. Columns were washed using 20 mM imidazole. Protein was eluted using 500 mM imidazole, dialyzed into 20mM Tris, 500 mM NaCl, 5mM $MgCl_2$, 2 mM BME, and 10% glycerol and stored at -20˚C. NahA protein was purified in a similar way except that NahA was induced for 1h at 30˚C and NahA expressing cells (JDE3138) were lysed, washed, and eluted in 250 mM NaCl instead of 500 mM.

## pGpp synthesis

pGpp was synthesized *in vitro* by purified NahA enzyme as described [14]. Briefly, 10 nM purified *B. subtilis* NahA was incubated with 30 mM pppGpp (Trilink Biotechnologies) in 40 mM

Tris-HCl (pH 7.5), 100 mM NaCl, 10 mM $MgCl_2$ at 37°C for 1 hour. Reactions were monitored for conversion of pppGpp to pGpp using thin layer chromatography on PEI-cellulose plates in 1.5 M $KH_2PO$ (pH 3.6). Nucleotides were visualized using short wave UV light. NahA enzyme was precipitated using ice cold acetone and nucleotides were stored at -20°C.

## SasB activity assays and TLC

SasB activity was assayed by measuring the amount of ppGpp generated similar to [5]. Briefly, 0.8 μM purified *B. subtilis* WT or F42A SasB was incubated with 0.5 μCi of [γ-$^{32}$P]-ATP (PerkinElmer) and 50 μM GDP in 20 mM Tris (pH 7.5), 500 mM NaCl, 5 mM $MgCl_2$, 2mM BME. SasB was allosterically activated using 12.5 μM pppGpp (Trilink Biotechnologies) and pGpp was added as noted. Reactions were performed in a total volume of 10 μL, and each reaction was incubated at 37°C for 1 min before being stopped using 5 μL of ice cold acetone. Conversion of ATP to ppGpp was visualized using thin layer chromatography on PEI-cellulose plates in 1.5 M $KH_2PO_4$ (pH 3.6). Plates were dried completely at RT and exposed for 5 min on a phosphor storage screen and visualized (GE Typhoon). ATP and ppGpp spot intensities were quantified using ImageJ.

## Supporting information

**S1 Fig. Determination of "OFF" cells using stationary phase cells.** Threshold for OPP "OFF" cells was determined as the fluorescence value (850 a.u.) that is higher than >95% of cells of OPP labeled wildtype *B. subtilis* during stationary phase across three independent experiments. **(A)** Three representative distributions of OPP labeled wildtype *B. subtilis*. Gray box shows cutoff for cells with low rates of protein synthesis ("OFF"). **(B)** Quantitation of % of population below the threshold determined as "OFF" in the three experiments in A (means ± SDs).
(PDF)

**S2 Fig. Late transition phase time point and application of "OFF" cutoff to transition phase cells. (A)** Growth curve of wildtype *B. subtilis* showing point ($OD_{600}$ ~0.685) where cells were labeled with OPP (dashed line). **(B)** Representative distribution of OPP labeled wildtype *B. subtilis*. Gray box shows cutoff for cells with low rates of protein synthesis ("OFF"). Threshold (850 a.u.) is the value higher than >95% of cells of wildtype *B. subtilis* labeled with OPP in stationary phase across three independent experiments. (see S1 Fig).
(PDF)

**S3 Fig. pGpp does not inhibit SasB basal activity. (Top)** representative TLC analysis of wildtype SasB activity in the absence of allosteric activation (no pppGpp added) and with increasing concentrations of pGpp (uM). **(Bottom)** ratio of ppGpp calculated using the formula, ppGpp/ATP + ppGpp. Statistical analysis (t-test) showed no significance (p > 0.05) between any reaction containing SasB whether or not pGpp was included. Statistical analysis was performed on three separate experiments (means ± SDs).
(PDF)

**S4 Fig. Effect of *nahA* on heterogeneity. (A, B)** Representative pictures and population distributions of OPP labeled **(A)** wildtype (JDB1772), **(B)** Δ*nahA* (JDB4095) strains during late transition phase.
(PDF)

**S1 Table. Plasmids used in this study.**
(PDF)

**S2 Table. Strains used in this study.**
(PDF)

**S3 Table. Oligonucleotides used in this study.**
(PDF)

## Author Contributions

**Conceptualization:** Simon Diez, Jonathan Dworkin.

**Funding acquisition:** Jonathan Dworkin.

**Investigation:** Simon Diez, Molly Hydorn, Abigail Whalen.

**Methodology:** Simon Diez.

**Project administration:** Jonathan Dworkin.

**Supervision:** Jonathan Dworkin.

**Writing – original draft:** Simon Diez, Jonathan Dworkin.

**Writing – review & editing:** Simon Diez, Jonathan Dworkin.

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
