## [Decision Letter · Decision Letter 0]

13 Jan 2022

Dear Dr Dworkin,

Thank you very much for submitting your Research Article entitled 'Crosstalk between guanosine nucleotides regulates cellular heterogeneity in protein synthesis during nutrient limitation' to PLOS Genetics.

The manuscript was fully evaluated at the editorial level and by independent peer reviewers. The reviewers appreciated the attention to an important problem, but noted some substantial concerns about the current manuscript. Reviewer 1 has raised concerns with attribution of specific Sas, Rel, and Nah activities to the reported phenotypes. This reviewer notes that providing experimental enzyme activity data would greatly strengthen the presented model.  In the absence of such data, the authors’ model would need to be revised. Along this same line, Reviewer 2 raises concerns about evidence for the presented model and attribution of specific sas activities and alarmone levels to the observed phenotypes. Reviewer 1 also raises questions about the determination of “OFF” cells that are important to address.

Based on the reviews, we will not be able to accept this version of the manuscript, but we would be willing to review a much-revised version. We cannot, of course, promise publication at that time.

If you decide to revise the manuscript for further consideration at PLOS Genetics, please aim to resubmit within the next 60 days, unless it will take extra time to address the concerns of the reviewers, in which case we would appreciate an expected resubmission date by email to plosgenetics@plos.org.

[LINK]

We are sorry that we cannot be more positive about your manuscript at this stage. Please do not hesitate to contact us if you have any concerns or questions.

Yours sincerely,

Sean Crosson

Associate Editor

PLOS Genetics

Lotte Søgaard-Andersen

Section Editor: Prokaryotic Genetics

PLOS Genetics

Reviewer's Responses to Questions

**Comments to the Authors:**

Reviewer #1: The paper by Diez et al that describes a possible basis for bimodal heterogeneity in translation levels in Bacillus subtilis cells in stationary phase. I have previously reviewed this manuscript for another journal and was disappointed to learn that few, if any, of the suggestions made by either reviewer were incorporated into this version of the manuscript. I thought these comments were generally constructive and have reiterated the key points from both referees here, because they remain fully pertinent.

One key issue is the oversimplification of the model whereby RelA produces pppGpp, SasA produces pGpp and SasB produces ppGpp. Each of these enzymes produces a mixture of products and different authors disagree on what the primary functions are in B. subtilis and closely related organisms. RelA and SasB have both been reported to produce mixtures of pppGpp and ppGpp (PMID: 6111556 ; 26124242). Both SasA and SasB have been reported to produce pGpp (PMID: 26124242 ; PMID: 22950019). Indeed, the primary reference cited in favor of the model that SasA produces pGpp and SasB produces ppGpp (ref 12; PMID: 22950019), was unable to distinguish between pGpp and ppGp by mass-spectrometery and favored the later as the most likely product of SasA in the discussion. This paper was not nearly as categorical as the current authors about which enzyme preferentially produces which product. Lastly, others have shown have shown that although SasA-expressing cells accumulate pGpp, most pGpp is from the conversion of ppGpp to pGpp by the hydrolase NahA (PMID: 32983059 ; 33097692). Ideally the authors should compare the activity of the 3 enzymes side-by-side with GTP, GDP and GMP as substrates. In the absence of such an experiment, the authors should be much more broad-minded about presentation of the model

The fluorescence cut-off value used to define cells as 'OFF' for translation was determined as the magnitude of OPP labeling of a stationary phase culture that captures 95% of the population (850 RFU). Unless I am misinterpreting Figure S1, the box showing the 'OFF' population represents <50% of the cells shown in this experiment, not 95% (I cut out the printed graph and weighed the peaks just to be sure). It also seems strange to assume that 95% of WT cells should be 'OFF' in stationary phase, when the study is all about population heterogeneity after all. Is this based on some previous determination? From visual inspection of the profiles shown throughout the paper, it seems that there is always a valley around 700-750 AU ; this might be a better choice for the cut-off.

Why were quartiles chosen for the analysis in Fig. 2C and D? Wouldn't it have been more logical just to divide into two groups, representing the two peaks? Is quartile 2 significantly different from 3 and 4? If the authors insist onf sticking with quartiles, please show where these fall on the graph using dotted lines. Also, please change the colors in panel 2B. The two shades of green are barely distinguishable.

Supplementary Figures S2, S3, and S5 should be deleted and the % 'OFF' cells +/- error and asterisks for significant differences be reported directly on the corresponding main figures in the text, to avoid having to go looking for this data in the supplementary section.

Please replace the term 'Ratio of ppGpp' on the Y-axis to ppGpp/(ppGpp+ATP).

Both referees had a problem with the use of the term epistatic towards the end of the results section. For me, sasB is epistatic to sasA and relA is epistatic to sasA, because the sasB phenotype dominates in the double sasAB mutant and the relA308 phenotype dominates in the sasA relA308 double mutant, ie cells are mostly 'ON' in both cases.

Minor points:

L56 Define RSH enzyme

L98 I am not sure how the authors arrive at the conclusion that 'This observation suggests that the SasA product pGpp does not directly inhibit translation as does ppGpp, but rather acts indirectly'. If anything, the phenotype of the sasA mutant could suggest that pGpp directly ACTIVATES translation, at least at face-value.

L202 IF2 is not the only translation-relevant target for ppGpp. Protein synthesis is also coupled to nutrient availability through the effect of ppGpp on ribosomal RNA and tRNA synthesis, and on ribosomal assembly factors.

Reviewer #2: In their interesting new Ms, Diez et al follow up their observation of a bimodal distribution of protein synthesis attenuation in a B. subtilis cell population, mediated e.g. by inhibiting IF2, under conditions when the the (p)ppGpp alarmone synthesis is induced (Diez et al. (2020) PNAS 117:15565). B. subtilis cells encode a bifunctional Rel synthetase, which can synthesize pGpp, ppGpp and pppGpp from GMP, GDP or GTP and ATP. Rel contains also a hydrolase domain, allowing the degradation of these alarmones.

In addition, there are also two small alarmone synthetases RelP (YwaC, SasA) & RelQ (YjbM, SasB) present in B. subtilis. The overlapping and also possibly distinct activities of the three different alarmones synthesized by the three different synthetases were recently investigated by different Labs and specifically for cellular pGpp in B. subtilis it was observed that a hydrolase NahA can generate pGpp from (p)ppGpp (Yang et al. (2020) Nat Comms 11:5388).

Here the authors investigated the possible involvement of the two small synthetases SasA and SasB in influencing the observed heterogeneity of protein synthesis detected by OPP labelling (Diez et al. (2020) PNAS 117:15565) (Fig 1A). Interestingly, when sasA was deleted, the translation was inhibited (Fig 1C) in most cells. However, a much higher portion of translating cells were observed in the population when sasB was deleted (Fig 1B).

The experiments in Fig 2 somehow imply that the cells with higher sasA transcription correlate more with the observation of higher translation. The authors argue that therefore differences in SasA expression might be more important for the observed changes in heterogeneity in that growthphase.

Interestingly a synthetase mutant (relAY308A) of Rel, which is probably in general the major source of (p)ppGpp displayed less bimodal distribution with mostly translating cells (Fig 3C), similar to ∆sasB cells (Fig 1B). And the same was true for a SasB variant F42A (Fig 3B), which is known not to be allosterically activated by pppGpp anymore. This allosteric activation results in higher pppGpp synthesis (Steinchen et al (2015) PNAS 112:13348). Both rel and sasB mutations result in more translating cell, consistent with an expected lower alarmone concentration in these strains.

The authors suggest that an allosteric regulation of SasB activity via SasA could explain these results. Since Tagami et al (2012 MicrobiolOpen 1:115) observe an increase of cellular pGpp, they suggest that the pGpp synthesized by SasA might interfere with SasB synthetase activity.

To test this hypothesis, the purified SasB and SasBF42A and show that raising amounts of pGpp somehow result in lowered synthesis of pppGpp. In the control experiment with SasBF42A this is not visible but the pppGpp synthesis is generally lower (Fig4).

Since both Rel and SasB influence SasA the double mutant ∆sasA∆sasB and ∆sasA rel Y308A should abolish the effect of ∆sasA, which can be confirmed experimentally as observed in Fig 5.

Based on these observations the authors suggest that the synthesis of pGpp by SasA negatively influences the Rel mediated activation of SasB and thereby the level of protein synthesis probably via IF2 in a bimodal manner (Fig6).

Comments

-To really give support to this model (Fig 6), it is probably important to measure the different alarmone levels (pGpp, ppGpp, pppGpp) in some of the presented key B. subtilis strains. Since the authors are able to separate the different populations by Fax, it might even be possible to determine this not only in the whole population but also in the respective sub-populations.

-However, it would already help to utilize in addition to the deletion strains of sasA and sasB synthetase defective SasA and SasB variants (e.g. SasAE154V SasBE139V). With such strains it would be easier to conclude that the observed phenotypes of the ∆sasA and/or ∆sasB strains really depend on the synthetase activity of SasA or SasB. And possible protein-protein interactions of these small synthetases, which might also support the observed genetic interaction, would be less perturbed in these strains.

-It is interesting to note that not only Tagami et al observed the in vivo synthesis of pGpp, when SasA was expressed in trans. Fung et al ((2020) Front Mic 11:2083) observed this too. But in addition, they observed a strong pleiotropic effect on the Bacillus physiology and metabolism, which also include the synthesis of other nucleotide second messengers.

Interestingly, Diez et al ((2020) PNAS 117:15565) show themselves in Fig S3 that a SasA induction in trans abolishes the protein synthesis in all the cells of such a population.

I think one could argue that pGpp synthesized by SasA could also directly or indirectly interfere with many cellular processes, which might also include translation.

Therefore, knowing the cellular concentration of pGpp would be important, since it is not clear at what cellular concentration the pGpp could start to influence SasB synthetase activity or the other observed pleiotropic processes.

-Fig 4 The presented in vitro experiments suggest that the pGpp might only diminish the allosteric activation, maybe by somehow competing with pppGpp? Or maybe pGpp competes with substrates?

A more comprehensive analysis and characterization of the SasB enzyme activity and the in vitro influence of pGpp on SasB, would allow to estimate for example determine affinities of pGpp, which would help to understand which cellular concentrations of pGpp have to be reached in vivo.

There seems to be no unit for the amount of pGpp added to the in vitro assay for the SasB activity. What are the concentrations and are they physiologically relevant?

- Fig S6 Both Yang et al and Fung et al observed that NahA is responsible for the generation of more than 80% of the cellular pGpp, possibly from (p)ppGpp synthesized by SasA. Therefore, it is very likely that in the absence of NahA the cellular pGpp concentration in such a strain is much reduced.

However, in Fig S6 the heterogeneity of protein synthesis of the ∆nahA strain looks like the wildtype strain. This suggests that in the absence of NahA the probably strongly reduced cellular pGpp does not make a difference, and might therefore not being majorly involved in generating the observed wild type like heterogeneity in translation.

other comments

Fig 2 The promoter activity might somehow correlate with the translated amount of SasA or SasB, however in the paper of Tagami et al they did a Western blot and showed that they were able to detect SasA only in low levels compared to SasB in the transition growth state.

l55, Intro RsgA might be a misleading example since in B. Subtilis RsgA does not bind ppGpp as shown in the cited paper (8) (Corrigan et al)

line 63-64 – „B. Subtilis RelA primarily generates pppGpp (...)” (11) This reference from Wendrich et al (1997) might not be the best fit, since at that time the two small synthetases were not known yet, and their additional influence could therefore not be considered in this paper at that time.

**Have all data underlying the figures and results presented in the manuscript been provided?**

Reviewer #1: None

Reviewer #2: Yes

PLOS authors have the option to publish the peer review history of their article (what does this mean?). If published, this will include your full peer review and any attached files.

Reviewer #1: No

Reviewer #2: No

---

## [Decision Letter · Decision Letter 1]

27 Mar 2022

Dear Dr Dworkin,

Thank you very much for submitting your Research Article entitled 'Crosstalk between guanosine nucleotides regulates cellular heterogeneity in protein synthesis during nutrient limitation' to PLOS Genetics.

This manuscript has been through a second round of review with your original set of reviewers. While some revisions your have made are appreciated, your reviewers again conclude that your conclusion/model are not supported by your data. Though you claim that Figure 6 is not a model, we agree with Reviewer 1 that it will be viewed as such by readers.

After editorial discussion, we agree that significant potential regulatory complexity is not accounted for in the discussion. Keeping Figure 6 in its present form requires some effort to measure nucleotide in the nahA deletion (even at the population level) under the relevant transition condition. If there's a reason why this is not possible, or a reason why such a measurement would be inconclusive, this would be helpful for the editors and reviewers to hear. Experiments that exclude the possibility that SasA or SasB have regulatory activities that do not require their synthase activities also are important if Figure 6 remains in its current form.

If you decide to revise the manuscript for further consideration at PLOS Genetics, please aim to resubmit within the next 60 days, unless it will take extra time to address the concerns of the reviewers, in which case we would appreciate an expected resubmission date by email to plosgenetics@plos.org.

[LINK]

We are sorry that we cannot be more positive about your manuscript at this stage. Please do not hesitate to contact us if you have any concerns or questions.

Yours sincerely,

Sean Crosson

Associate Editor

PLOS Genetics

Lotte Søgaard-Andersen

Section Editor: Prokaryotic Genetics

PLOS Genetics

Reviewer's Responses to Questions

**Comments to the Authors:**

Reviewer #1: The revised ms by Diez et al was again disappointing, with little effort to provide any of the experimental support requested by either reviewer. My main issue still is the oversimplification of the model (although the authors say that Fig. 6 is not a model, most readers will regard it as such). Even though the different (pp)pGpp synthetases may generate preferential products under specific circumstances, they are nonetheless mixed at best, and in some cases not yet completely determined. Despite a minor text modification, IF2 comes across as the only translation target of (p)ppGpp, when several other components of the translational apparatus, including rRNA and tRNA synthesis, are well known to be (p)ppGpp targets. The paper globally, and the schematic in Fig. 6, need be redone to incorporate other possibilities. For example, when a mixture of products is known to be produced, the favored one could have a larger font in Fig. 6. pGpp could have an asterisk to indicate that the inhibitory effect was shown directly, but ppGp should also be shown. rRNA and tRNA should be mentioned along with IF2 as targets of (pp)pGpp.

The product of SasA is either pGpp or ppGp, but more likely ppGp, according to the authors of the cited paper PMID 2950019. For reasons that are unclear, the authors propose to call this mixture pGpp (which I find a little disingenuous), and then only test pGpp (and not ppGp) in Fig. 4 as a competitive inhibitor of the allosteric activation of SasB by pppGpp. The mixture should be referred to as pGpp/ppGp.

As referee 2 pointed out, both Yang et al and Fung et al observed that NahA is responsible for the generation of more than 80% of the cellular pGpp, possibly from (p)ppGpp synthesized by SasA. Therefore, it is very likely that in the absence of NahA the cellular pGpp concentration in such a strain is much reduced and the fact the nahA deletion strain behaves like the wt is not coherent with the model presented. To explain this discrepancy, the authors make the not very convincing suggestion that this may not be the case in late transition phase, but do not provide any evidence to support this. In the absence of experimental evidence, this for me is a potential symptom of an over-simplified model.

I agree with referee 2 that it is important to measure the different nucleotides present in the different mutants under the experimental conditions tested to provide support for the model, even if it is only technically feasible at the population level.

Minor points:

Fig. S1 should be described in the text. The sentences provided L88-91 reference Fig. S2 and still refer to 95% of entire population in late transition phase, when this is clearly not the case.

It is still not clear to me how the quartiles were designated in Figure 2. Only two seem to be delineated in Figure 2A and B. Labelling the quartiles would help. Is only the late transition peak considered? If so how was the growth phase of the two peaks determined?

Abstract (L29) There is one ‘p’ too many in (pp)ppGpp

Reviewer #2: I attached a PDF file of the ReReview text.

**Have all data underlying the figures and results presented in the manuscript been provided?**

Reviewer #1: Yes

Reviewer #2: Yes

PLOS authors have the option to publish the peer review history of their article (what does this mean?). If published, this will include your full peer review and any attached files.

Reviewer #1: No

Reviewer #2: No

---

## [Editor Report · Decision Letter 2]

24 Apr 2022

Dear Dr Dworkin,

We are pleased to inform you that your manuscript entitled "Crosstalk between guanosine nucleotides regulates cellular heterogeneity in protein synthesis during nutrient limitation" has been editorially accepted for publication in PLOS Genetics. Congratulations!

Yours sincerely,

Sean Crosson

Associate Editor

PLOS Genetics

Lotte Søgaard-Andersen

Section Editor: Prokaryotic Genetics

PLOS Genetics

Comments from the reviewers (if applicable):

**Data Deposition**

http://datadryad.org/submit?journalID=pgenetics&manu=PGENETICS-D-21-01528R2

**Press Queries**

---

## [Editor Report · Acceptance letter]

16 May 2022

PGENETICS-D-21-01528R2 

Crosstalk between guanosine nucleotides regulates cellular heterogeneity in protein synthesis during nutrient limitation 

Dear Dr Dworkin, 

We are pleased to inform you that your manuscript entitled "Crosstalk between guanosine nucleotides regulates cellular heterogeneity in protein synthesis during nutrient limitation" has been formally accepted for publication in PLOS Genetics! Your manuscript is now with our production department and you will be notified of the publication date in due course.

With kind regards,

Livia Horvath

PLOS Genetics

On behalf of:
